# Deep Combinatorial Aggregation

**Yuesong Shen** [1,2]     **Daniel Cremers** [1,2]
[1] Technical University of Munich, Germany
[2] Munich Center for Machine Learning, Germany
{yuesong.shen, cremers}@tum.de

## Abstract

Neural networks are known to produce poor uncertainty estimations, and a variety of approaches have been proposed to remedy this issue. This includes deep ensemble, a simple and effective method that achieves state-of-the-art results for uncertainty-aware learning tasks. In this work, we explore a combinatorial generalization of deep ensemble called deep combinatorial aggregation (DCA). DCA creates multiple instances of network components and aggregates their combinations to produce diversified model proposals and predictions. DCA components can be defined at different levels of granularity. And we discovered that coarse-grain DCAs can outperform deep ensemble for uncertainty-aware learning both in terms of predictive performance and uncertainty estimation. For fine-grain DCAs, we discover that an average parameterization approach named deep combinatorial weight averaging (DCWA) can improve the baseline training. It is on par with stochastic weight averaging (SWA) but does not require any custom training schedule or adaptation of BatchNorm layers. Furthermore, we propose a consistency enforcing loss that helps the training of DCWA and modelwise DCA. We experiment on in-domain, distributional shift, and out-of-distribution image classification tasks, and empirically confirm the effectiveness of DCWA and DCA approaches. [1]

## 1   Introduction

Deep learning has achieved groundbreaking progress and neural networks are now widely used in various domains [24]. However, they are known to produce poor uncertainty estimations [9, 37, 10], which can be problematic for challenges like safety-critical applications [17, 26] or active learning [35]. Numerous approaches have been proposed to tackle this issue, among which an effective yet simple method is deep ensemble [23]. Deep ensemble yields state-of-the-art results for uncertainty aware learning [37, 10], and it does not require elaborate architectural design and hyperparameter search. However, while it aggregates multiple separately trained models, it can neither generate new samples from the posterior to obtain more diverse predictions, nor produce a summarizing average model which improves on the individual models in the ensemble.

Motivated by the success of deep ensemble, in this paper, we propose deep combinatorial aggregation (DCA), which generalizes via a combinatorial perspective: given the hierarchical structure of neural networks, we explore the idea of ensembling components of the network architecture and combining them to form an enriched collection of model proposals. DCA inherits the simplicity and effectiveness of deep ensemble, and additionally leads to several new possibilities: Apart from generating a diversified set of model proposals, we discover that fine-grain DCA can lead to a new average proposal via deep combinatorial weight averaging (DCWA). Furthermore, DCA training can benefit from a consistency enforcing loss, which can produce DCA models that surpass standard deep ensemble both in classification performance and uncertainty estimation.

---

[1] Source code is available at `https://github.com/tum-vision/dca`.

36th Conference on Neural Information Processing Systems (NeurIPS 2022).

## 1.1 Related work

Improving not only the predictive performance but also the uncertainty estimation of neural networks has been a core objective of Bayesian deep learning, where an abundance of prior work exists. This includes methods based on variational inference and weight perturbation such as Bayes by backprop [8, 2] and its variants [41, 4], Bayesian interpretation of dropout including MC dropout [6] and variational dropout [1, 30], expectation propagation which leads to probabilistic backpropagation [14], Markov chain Monte Carlo (MCMC) with methods like stochastic gradient Langevin dynamics (SGLD) [40] and stochastic gradient Hamiltonian Monte Carlo (SGHMC) [3], stochastic gradient descent (SGD) as approximate MCMC which results in SWAG [28], as well as Bayesian noisy optimizers such as variational online Gauss-Newton (VOGN) [18] and approaches using Laplace approximation [20, 33]. Beyond the Bayesian formulation, methods like post-hoc calibration [9, 39] readjust trained networks to produce more calibrated predictions, while approaches like evidential deep learning [34] also make use of ideas like subjective logic.

Most relevant to our work is the deep ensemble method [23], which aggregates multiple independently trained network models with different initial parameters. Deep ensemble has been shown to produce state-of-the-art results for uncertainty estimation [37, 10]. It can be combined with methods like MC dropout [5] and SWAG [44], or extended to the hyperparemeter space [43]. Several variants have also been proposed, which often aim at providing more computationally or memory-efficient alternatives. This includes snapshot ensemble [15], BatchEnsemble [42], fast geometric ensembling [7], TreeNet [25], *etc*. In contrast, this work aims at exploring a combinatorial generalization of deep ensemble to obtain new features and improve the performance of uncertainty estimation.

Lastly, our proposed DCWA method provides an alternative weight averaging scheme comparable to stochastic weight averaging (SWA) [16].

## 1.2 Contributions

The main contributions of this paper are the following:

- We propose deep combinatorial aggregation, a combinatorial generalization of deep ensemble that can produce more diverse model proposals and predictions.
- We explore DCA at different levels of granularity and propose deep combinatorial weight averaging (DCWA) for fine-grain DCA models. It produces a new average model that improves on standard training and is competitive w.r.t. alternatives like stochastic weight averaging (SWA) [16].
- We introduce a consistency enforcing loss adapted for DCA training. It strengthens the predictive consistency of DCA model proposals and consistently improves the performance of DCA and DCWA models.
- We conduct experiments on in-domain, distributional shift, and out-of-distribution image classification tasks, which validate our analysis and demonstrate the effectiveness of DCA for uncertainty-aware learning.

## 2 Deep combinatorial aggregation (DCA)

In this section, we introduce the *deep combinatorial aggregation (DCA)* method. To simplify our discussion, we start with a layerwise setting and assume that our base model is a neural network with $L$ layers. This setting can easily be generalized to other DCA variants discussed in Section 2.3.

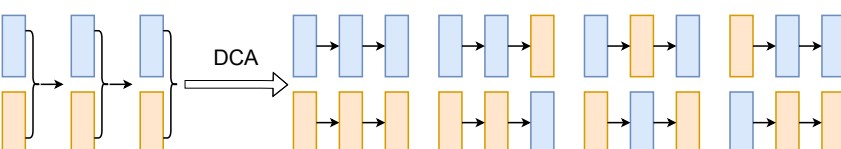

Figure 1: Illustration of layerwise deep combinatorial aggregation using a three-layer neural network: two sets of DCA parameter instances result in $2^3 = 8$ model proposals. A random proposal is chosen for each forward pass during both training and test time to generate diverse prediction samples.

## 2.1 Methodology

The main idea of layerwise deep combinatorial aggregation is straightforward: while deep ensemble [23] creates multiple copies of the entire model with different initializations, layerwise DCA instead creates multiple instances for each layer in the model. Randomly selected instances from network layers can be combined to form a variety of model proposals. As shown in Figure 1, this results in an exponential number of proposals w.r.t. the network depth, since $n$ sets of layer instances lead to $n^L$ total network proposals.

To ensure consistency among the instance combinations, all layer instances are jointly trained: during each feed-forward, a random instance choice from each layer is made to construct a model proposal, and the parameters belonging to the selected layer instances are then updated via backpropagation. This differs from deep ensemble [23] where model copies are independently trained. In Appendix A we provide a pseudo-code for layerwise DCA training.

During inference time, to obtain uncertainty-aware predictions, one can simply sample multiple DCA model proposals and aggregate their predictions.

## 2.2 Understanding DCA

While DCA is a simple and intuitive procedure, it is helpful to conduct a more in-depth theoretical analysis to understand the assumptions it implicitly makes and their implications.

Being able to freely combine samples of layer parameters assumes that they follow mutually independent distributions. This translates to the assumption that the weight posterior $p(\boldsymbol{\theta}|y, x)$ should be layerwise decomposable:

$$p(\boldsymbol{\theta}|y, x) \propto \prod_{l=1}^{L} \phi^l(\theta^l). \tag{1}$$

Is this a valid assumption? Not exactly. Admittedly, the posterior is proportional to the product of prior $p(\boldsymbol{\theta})$ and likelihood $p(y|x, \boldsymbol{\theta})$ (i.e. $p(\boldsymbol{\theta}|y, x) \propto p(\boldsymbol{\theta})p(y|x, \boldsymbol{\theta})$), and the weight prior $p(\boldsymbol{\theta})$ often satisfies a layerwise independence assumption, e.g., commonly used weight decay is equivalent to Gaussian prior with constant diagonal covariance matrix. However, the likelihood term $p(y|x, \boldsymbol{\theta})$ is in general not decomposable. In fact, following the directed graphical model [21] interpretation of standard neural networks [36], the base network represents the overall distribution $p(h^{1:L-1}, y|x, \boldsymbol{\theta})$ where $h^l$ represents hidden neurons in layer $l$. Interestingly, $p(h^{1:L-1}, y|x, \boldsymbol{\theta})$ is actually layerwise decomposable itself

$$p(h^{1:L-1}, y|x, \boldsymbol{\theta}) = \prod_{l=1}^{L} p(h^l|h^{l-1}, \theta^l), \quad (x, y := h^0, h^L). \tag{2}$$

Nevertheless, the likelihood term $p(y|x, \boldsymbol{\theta})$ requires marginalization of all hidden neurons $h^{1:L-1}$

$$p(y|x, \boldsymbol{\theta}) = \int_{h^{1:L-1}} p(h^{1:L-1}, y|x, \boldsymbol{\theta})dh^{1:L-1}. \tag{3}$$

This entangles the layer parameters, and $p(y|x, \boldsymbol{\theta})$ is no longer decomposable in general.

Thus approximations are made when we perform layerwise DCA, and experiments show that this indeed results in some performance penalty. The above analysis also implies that weight aggregation at a coarser multilayer level could alleviate the issue. This is also confirmed by our empirical findings (cf. Section 5.3).

## 2.3 Granularity of aggregation

The analysis in Section 2.2 raises an interesting concern about doing deep combinatorial aggregation at different levels of granularity: From finest to coarsest, DCA can be defined at neuronwise, layerwise, multilayer, or modelwise levels. For convolutional neural networks, due to weight-sharing along the spatial dimensions, DCA can work with channel components instead of neurons. Note that modelwise DCA is similar to deep ensemble, except that in each epoch DCA model copies are trained with distinct subsets of training data which results from the random component selection, and it can benefit from consistency enforcing loss introduced in Section 4.

In general, DCA with finer granularity generates a greater amount of model proposals and more varied predictions. However, the DCA components are more tightly coupled, deviating more significantly from the assumption of decomposable posterior, which could lead to worse performance. This results in a tradeoff between performance and prediction variety. Alternatively, one can also enrich the set of model proposals by using more DCA instances, at the cost of a higher computational budget. This can also lead to improved performance (*cf*. Section 5.3).

Among the broad range of granularities, there exists a notable dichotomy between (sub)layerwise DCA and multilayered DCA. This comes from the fact that component instances from multilayered DCA variants can have similar behaviors but dissimilar weights. Neural networks admit a large number of equivalent reparameterizations via the reordering of hidden layer neurons. While the joint training of DCA enforces consistency among different DCA components of the network model, it does not prevent equivalent reordering of hidden neurons inside multilayer components. This issue does not occur for (sub)layerwise DCA cases. To make the distinctions, we refer to (sub)layerwise cases as fine-grain DCA and multilayered cases as coarse-grain DCA.

# 3    Deep combinatorial weight averaging (DCWA) for fine-grain aggregation

For fine-grain DCA models, it turns out that averaging the learned weights of DCA components leads to an improved parameterization of the base model. We discuss this procedure here in detail.

**Deep combinatorial weight averaging**    Since component instances of a fine-grain DCA model have compatible weights after the joint training, it is sensible to consider their mean value. This produces a new average parameterization for the base network model. We refer to this process as *deep combinatorial weight averaging (DCWA)*.

Consider as an example the layerwise DCA model for a base neural network with $l$ layers. After the joint training of $n$ sets of DCA layer instances parameterized by $\mathbf{\Theta} = (\theta_{1:n}^1, \ldots, \theta_{1:n}^L)$, DCWA simply computes the average parameterization $\bar{\boldsymbol{\theta}} = (\bar{\theta}^1, \ldots, \bar{\theta}^L)$ for the base network model, where for each layer $l$, we have $\bar{\theta}^l = \frac{1}{n} \sum_{i=1}^n \theta_i^l$.

Experiments show that DCWA achieves comparable test accuracy w.r.t. corresponding DCA predictions (*cf*. Section 5.3). It also consistently outperforms the standard training of the base network, and delivers comparable results w.r.t. to SWA [16] (*cf*. Section 5.1).

**Comparison to SWA**    It is interesting to compare DCWA with SWA [16] since they are both weight averaging schemes that improve the standard training. This said, DCWA and SWA are based on different principles: SWA averages over the SGD trajectory while DCA relies on combining component aggregations. In practice, SWA requires custom learning rate scheduling and careful choice of end learning rate. Also, SWA requires an additional Batch Normalization update [16] to produce good predictions, which leads to an extra overhead. In contrast, DCWA does not have any of these issues and is simple to implement and deploy.

# 4    Consistency enforcing loss for DCA and DCWA

Through component combination, DCA is able to produce a combinatorial amount of model proposals. However, during the joint training, each DCA component receives gradient updates from different model proposals. These updates can be inconsistent, which can lead to suboptimal training of the DCA model. To remedy this issue, we propose in this section a consistency enforcing loss to encourage consistency among DCA model proposals.

**Consistency enforcing loss**    To promote consistency among DCA model proposals, we encourage DCA predictions agree with both the ground-truth and the predictions from other model proposals. To achieve this, given an input $x$ with ground-truth $y$ and a DCA model proposal parameterized by $\hat{\boldsymbol{\theta}}$, instead of minimizing the negative log-likelihood $\ell_{NLL}(x, y; \hat{\boldsymbol{\theta}}) = -\log p(y|x; \hat{\boldsymbol{\theta}})$, the consistency enforcing loss includes an additional KL divergence term between the predictive output probability $p(y|x; \hat{\boldsymbol{\theta}})$ and a reference output probability $\tilde{p}$:

$$\ell(x, y, \tilde{p}; \hat{\boldsymbol{\theta}}) = -\log p(y|x; \hat{\boldsymbol{\theta}}) + D_{KL}(\tilde{p} \| p; \hat{\boldsymbol{\theta}}). \tag{4}$$

Here the reference output probability $\tilde{p}$ should be chosen to reflect the predictions from other model proposals. In practice, we perform multiple feed-forward steps on the same input $x$ using randomly chosen model proposals $(\hat{\boldsymbol{\theta}}^{(i)})_{1 \leq i \leq s}$, and the $i$-th consistency enforcing loss $\ell^{(i)}$ simply uses the output probability prediction $p^{(i-1)}$ from the previous step as reference distribution: $\ell^{(i)} = \ell(x, y, p^{(i-1)}; \hat{\boldsymbol{\theta}}^{(i)})$. Note that the output predictions are reused and one additional feed-forward pass is needed to compute a first reference distribution $p^{(0)}$. The gradient updates from the multiple feed-forward steps can be accumulated together for a single optimization step on the global DCA parameter set $\boldsymbol{\Theta}$.

Also, observe that the consistency enforcing loss uses a forward KL divergence $D_{KL}(\tilde{p}\|p)$ to regularize the predictive output distribution $p$. The forward KL divergence $D_{KL}(\tilde{p}\|p)$ is "zero-avoiding" and allows $p$ to be flat and uncertain. This is helpful for the output predictive distribution $p$ to find a good compromise between the reference prediction $\tilde{p}$ and the ground-truth $y$, especially in cases where they strongly disagree with each other.

**Relation to existing approaches** KL divergence as a regularization term in loss function appears in many contexts: for Bayesian neural networks [8, 2] it is used to regularize weight posterior; in variational autoencoders (VAE) [19] it regularizes the latent representation. Applying KL divergence on output distributions has been proposed by Zhang et al. [46] for knowledge distillation, and Song and Chai [38] for collaborative learning. However, to the best of our knowledge, this has not been extensively explored in the context of uncertainty-aware learning. Also, existing approaches [46, 38] require output predictions from all other models, which is impractical for DCA with a combinatorial amount of model proposals. In comparison, our proposed consistency enforcing loss is efficient and synergizes well with DCA training.

**Consistency *vs*. diversity** While strengthening the predictive consistency can help the DCA parameter training, it has the risk of reducing the diversity of model predictions. Since model diversity has been found beneficial for ensemble approaches [43, 45, 29], the consistency enforcing loss has a mixed effect on uncertainty estimation and could be advantageous in some cases. In practice (*cf*. Section 5.3), we observe that the consistency enforcing loss boosts the predictive performance of DCWA models and improves the uncertainty estimation of modelwise DCA.

To further analyze the consistency enforcing loss, we empirically study its effect on the classification performance of individual DCA model proposals. The results are collected in Appendix C. We observe that the consistency enforcing loss improves the prediction results of individual models but reduces their diversity.

## 5 Experiments

To empirically evaluate the effectiveness of the proposed DCA and DCWA models, we conduct a series of experiments: we benchmark the predictive performance of DCWA in Section 5.1 and evaluate the uncertainty estimation of DCA models in Section 5.2. Also, ablation studies on DCA variants are included in Section 5.3 to further analyze several design aspects. For all experiments we use a common preactivation ResNet-20 architecture [11] as the base network, and five independent runs are executed to produce the final results. Our implementation uses PyTorch [32] and can run on a single modern GPU with 10Gb VRAM.

### 5.1 Image classification using DCWA

Table 1: Image classification results on CIFAR-10 and SVHN. DCWA has comparable performance w.r.t. SWA and is simple to use. Both SWA and DCWA outperform the standard baseline training.

| | CIFAR-10 | | SVHN | |
| --- | --- | --- | --- | --- |
| | Accuracy $\uparrow$ | NLL $\downarrow$ | Accuracy $\uparrow$ | NLL $\downarrow$ |
| Standard | $0.9264 \pm 0.0020$ | $0.2507 \pm 0.0078$ | $0.9590 \pm 0.0012$ | $0.1786 \pm 0.0038$ |
| SWA | $0.9327 \pm 0.0016$ | $0.2167 \pm 0.0069$ | $0.9610 \pm 0.0010$ | $0.1809 \pm 0.0016$ |
| DCWA | $0.9337 \pm 0.0022$ | $0.2355 \pm 0.0101$ | $0.9606 \pm 0.0024$ | $0.1713 \pm 0.0060$ |

Firstly, we evaluate the performance of the DCWA approach. For this we use CIFAR-10 [22] and SVHN [31] datasets and compare against the standard training and SWA [16] as baselines. DCWA models are obtained from training layerwise DCA models with five copies and consistency enforcing loss is used. SWA models are trained following the settings from Izmailov et al. [16]. Test accuracy and negative log-likelihood (NLL) on both datasets are reported in Table 1.

We observe that DCWA and SWA have comparable results, and both of them outperform the standard training baseline. However, DCWA has the appealing practical advantages over SWA of not requiring custom learning rate scheduling nor readjustment of batch normalization layers.

## 5.2 DCA for uncertainty-aware learning

We now focus on comparing the uncertainty estimation of DCA models against a selection of practical uncertainty-aware learning methods, including MC dropout [6], SWAG [28], and deep ensemble [23]. Also, results from the standard training of the base network are reported for reference. These baselines are compared with trunkwise[2] and modelwise DCA models. Following the ablation study results in Table 3, consistency enforcing loss is used with modelwise DCA models while standard negative log-likelihood loss is used to train trunkwise DCA models for best performance.

To evaluate the quality of uncertainty estimation, we conduct a series of experiments on in-domain, distributional shift, and out-of-distribution (OOD) image classification problems. We consider common metrics for uncertainty quantification, including negative log-likelihood (NLL), Brier score, and expected calibration error (ECE) [9]. For OOD experiments, we follow the settings of Liang et al. [27] and plot the ROC-curve, and report quantitative metrics such as FPR95, detection error, AUPR-in, AUPR-out, and AUROC in Appendix B.3.

Overall, we observe that modelwise DCA produces the best results both in terms of predictive performance and uncertainty estimation. It is the method of choice if the performance of uncertainty-aware learning is the only concern. Trunkwise DCA is overall competitive with deep ensemble and moreover it produces more diverse model proposals. It offers a trade-off for more varied predictions.

**In-domain comparison**  We start by training the baselines and DCA models for the standard in-domain image classification problems on the CIFAR-10 [22] and SVHN [31] datasets. For all training we use SGD with momentum $0.9$. We use a drop rate of $0.1$ for MC dropout, and for SWAG we follow the settings from Maddox et al. [28]. Deep ensemble is performed on five separate base networks. And trunkwise and modelwise DCA models also use five copies. The results on CIFAR-10 are summarized in Table 2 and SVHN results are in Appendix B.1.

Table 2: In-domain image classification results on CIFAR-10. In general, modelwise DCA has the best predictive performance and uncertainty estimation, while trunkwise DCA has slightly better results w.r.t. deep ensemble while producing a richer set of model proposals.

|  | Accuracy ↑ | NLL ↓ | ECE ↓ | Brier ↓ |
|---|---|---|---|---|
| Standard | $0.9264 \pm 0.0020$ | $0.2507 \pm 0.0078$ | $0.0301 \pm 0.0020$ | $0.1132 \pm 0.0030$ |
| MC dropout | $0.9281 \pm 0.0015$ | $0.2186 \pm 0.0034$ | $0.0121 \pm 0.0013$ | $0.1057 \pm 0.0021$ |
| SWAG | $0.9329 \pm 0.0018$ | $0.2016 \pm 0.0045$ | $0.0149 \pm 0.0023$ | $0.0994 \pm 0.0020$ |
| Deep ensemble | $0.9434 \pm 0.0017$ | $0.1746 \pm 0.0028$ | $\underline{0.0085 \pm 0.0011}$ | $0.0834 \pm 0.0013$ |
| Trunk. DCA | $0.9454 \pm 0.0008$ | $0.1676 \pm 0.0018$ | $\underline{0.0080 \pm 0.0011}$ | $0.0809 \pm 0.0011$ |
| Model. DCA | $\underline{0.9481 \pm 0.0008}$ | $\underline{0.1550 \pm 0.0008}$ | $\underline{0.0084 \pm 0.0009}$ | $\underline{0.0762 \pm 0.0007}$ |

In general, modelwise DCA obtains the best performance both in terms of classification accuracy and uncertainty estimation. Also, trunkwise DCA slightly outperforms deep ensemble in this case. SVHN results in Appendix B.1 also confirm these observations.

**Distributional shift experiments**  In-domain experiments alone are not sufficient to fully characterize the quality of uncertainty estimation, since they do not reflect the case when there is a mismatch between the training data and the test set. This calls for additional distributional shift experiments,

---

[2]A trunk consists of a (residual) block with a non-identity main branch together with subsequent residual blocks having an identity main branch, following the graphical model interpretation of Shen and Cremers [36].

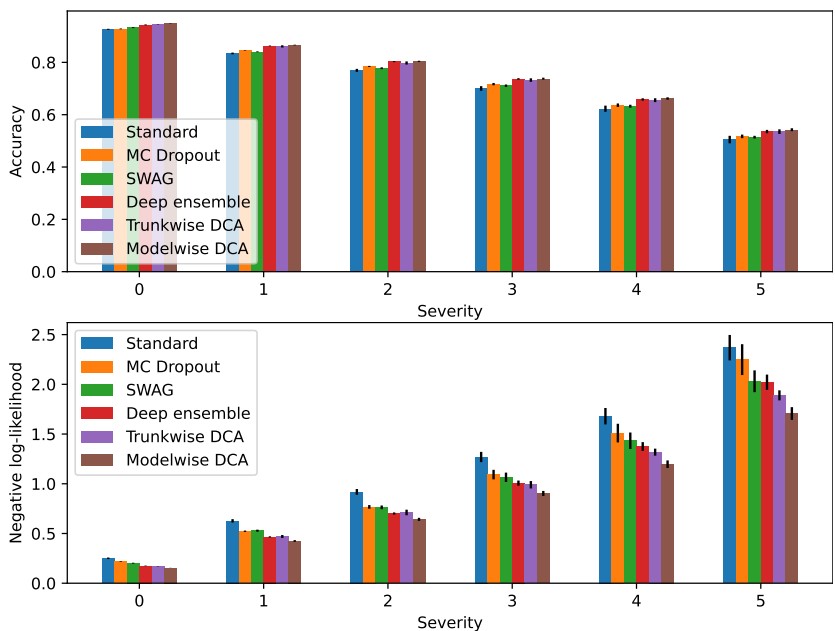

Figure 2: Distributional shift image classification results on CIFAR-10-C with various degrees of corruption severities. Again, modelwsie DCA has the best overall performance. Severity 0 corresponds to the in-domain case.

which gradually transform the in-domain test set to make it increasingly different than the training data. In our case, we use the CIFAR-10-C [12] dataset, which applies a wide range of corruptions to the original CIFAR-10 test set. CIFAR-10-C quantizes corruptions into five severity levels where higher levels generate stronger corruptions and are more dissimilar to the in-domain test set. We use the models previously trained on CIFAR-10 in-domain training data, and evaluate their performance on CIFAR-10-C separately for each severity level. Figure 2 summarizes the results.

Similar to the in-domain scenario, modelwise DCA has the best results overall. Trunkwise DCA has comparable test accuracies w.r.t. deep ensemble. It tends to outperform deep ensemble in terms of negative log-likelihood, however, it is worse in terms of Brier score and ECE, as indicated by the additional plots in Appendix B.2.

**Out-of-distribution results** Additionally, we complement the distributional shift experiments with out-of-distribution image detection [13]. The goal is to distinguish in-domain test samples from outliers that are drawn from a different dataset. Thus we have a binary classification setting. For our case, we reuse again the models trained on in-domain CIFAR-10 training data. Samples from the CIFAR-10 test set are regarded as in-domain positive samples while test samples from SVHN are treated as negative outliers. We show the ROC curves for DCA models and their baselines in Figure 3. Quantitative metrics following Liang et al. [27], which include FPR95, detection error, AUPR-in, AUPR-out and AUROC, are reported in Appendix B.3.

Again, we observe that modelwise DCA achieves the best OOD performance. Trunkwise DCA has the second-best performance and outperforms deep ensemble and SWAG. Inter-

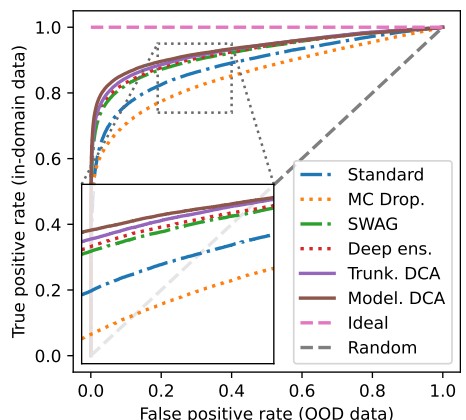

Figure 3: ROC curves with CIFAR-10 as the in-domain dataset and SVHN as the OOD dataset. Modelwise DCA shows the best performance for OOD detection, followed by trunkwise DCA.

estingly, MC dropout turns out to perform poorly in the OOD case and is even worse than the standard training baseline. The quantitative results in Appendix B.3 agree with the ROC curve.

## 5.3 Ablation studies

We wrap up the experiment section with a series of ablation studies on variants of DCA models. We consider three main aspects: the effect of granularity, the usage of consistency enforcing loss during training, and the influence of component instance count on DCA predictive results. We use the in-domain CIFAR-10 classification task to compare the performance of different DCA variants.

**Effect of granularity**    To understand the effect of DCA component granularity on the performance of DCA models, we define five DCA model variants with different component granularities. This includes DCA models whose components are channels, layers, residual blocks, residual trunks, and entire models. All DCA model variants are constructed with five sets of component instances and trained with the consistency enforcing loss (NLL loss leads to similar observations). We summarize the results of these DCA variants and their DCWA models in Figure 4.

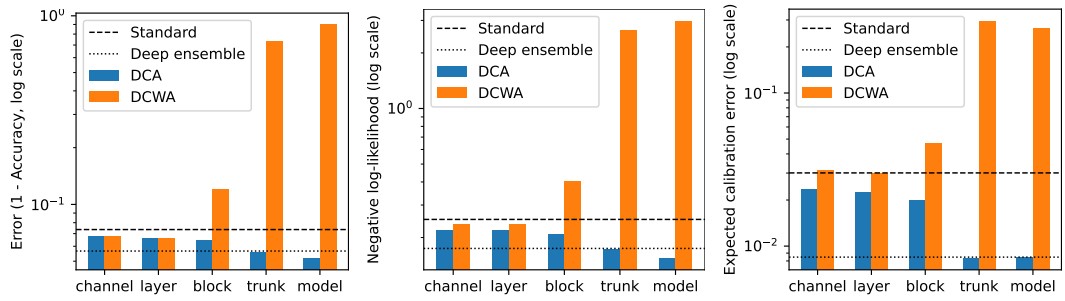

Figure 4: Classification errors ($= 1-$accuracy, lower is better), NLLs, and ECEs of DCA (blue) / DCWA (orange) models with different component granularities on in-domain CIFAR-10 classification. As references, the results of standard baseline training (dashed lines), as well as deep ensemble (dotted lines), are also provided. The metrics are all plotted in log scale.

Several interesting observations can be drawn from this experiment: For DCA models, coarser component granularity generally leads to better results. This agrees with the analysis in Section 2.2. For DCWA models we observe a clear distinction between fine-grain (channel and layer component DCAs) and coarse-grain (block, trunk, and model component DCAs) variants: fine-grain DCWA variants improve on the standard baseline, while coarse-grain DCWA variants result in considerably worse performance than the standard baseline. This also matches our analysis in Section 2.3. Interestingly, the DCWA model with block component is still able to yield significantly better than random (*i.e.* 10%) accuracy. In this case, reparameterization mismatch only happens on the side branches of the residual blocks, and the trained weights on the main branch are still able to produce meaningful predictions despite the erroneous parameterization on the residual side branches.

Table 3: Comparison of DCA training using negative log-likelihood (NLL) loss and consistency enforcing loss (CEL) for (layerwise) DCWA, trunkwise DCA, and modelwise DCA models.

|  | Accuracy ↑ | NLL ↓ | ECE ↓ |
|---|---|---|---|
| DCWA (NLL) | $0.9314 \pm 0.0017$ | $0.2651 \pm 0.0066$ | $0.0365 \pm 0.0014$ |
| DCWA (CEL) | $0.9337 \pm 0.0022$ | $0.2355 \pm 0.0101$ | $0.0301 \pm 0.0014$ |
| Trunkwise DCA (NLL) | $0.9454 \pm 0.0008$ | $0.1676 \pm 0.0018$ | $0.0080 \pm 0.0011$ |
| Trunkwise DCA (CEL) | $0.9441 \pm 0.0020$ | $0.1728 \pm 0.0043$ | $0.0084 \pm 0.0017$ |
| Modelwise DCA (NLL) | $0.9495 \pm 0.0008$ | $0.1601 \pm 0.0006$ | $0.0107 \pm 0.0004$ |
| Modelwise DCA (CEL) | $0.9481 \pm 0.0008$ | $0.1550 \pm 0.0008$ | $0.0084 \pm 0.0009$ |

**Consistency enforcing loss**    To analyze the influence of consistency enforcing loss for DCA training, we compare DCWA, trunkwise DCA, and modelwise DCA models which are trained with

negative log-likelihood loss with those trained with consistency enforcing loss. We summarize the results on CIFAR-10 in Table 3.

We observe that the consistency enforcing loss improves the overall performance of the DCWA model. It is also beneficial for the uncertainty estimation of the modelwise DCA model, both in terms of NLL and ECE. On the other hand, negative log-likelihood loss is the better choice for the trunkwise DCA model. In Appendix B.4 we provide additional results for other DCA variants.

**Number of component instances**   Finally, we investigate the influence of DCA instance count. To properly train all the DCA component instances jointly, we find out that it is beneficial to multiply the training epochs and per-minibatch gradient backpropagations accordingly. For example, in our experiments, we schedule 200 epochs for standard baseline training. Then, for training a DCA model with three sets of component instances we use 600 epochs, and accumulate for each minibatch an average gradient over three backpropagations on randomly selected DCA proposals for parameter update. We train a series of layerwise DCA models with two, three, four, and five sets of component instances, and collect their results in Figure 5.

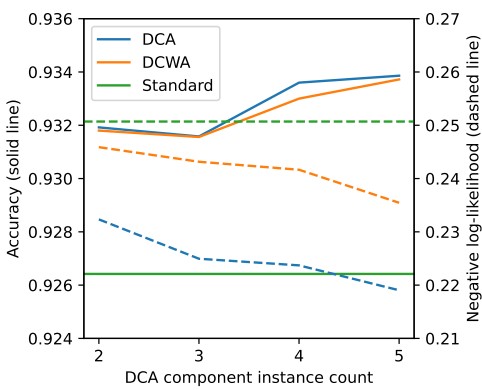

Figure 5: Accuracy (solid lines) and negative log-likelihood (dashed lines) of layerwise DCA models (blue) and their corresponding DCWA models (orange) with two, three, four and five sets of DCA components. The results of standard baseline training (green) are also provided for reference.

In general, we observe that more DCA instances lead to better results (higher accuracy and lower NLL loss in Figure 5). Therefore, DCA provides a reasonable trade-off between the computational footprint and the predictive power. Moreover, we see that a minimal DCA model with two sets of components can already significantly improve the performance compared to the standard base network training case.

# 6   Discussion

In this work, we propose deep combinatorial aggregation, a generalization of deep ensemble that has also a simple setup and, unlike the choice of dropout rate for MC dropout or the custom learning rate scheduling of SWAG, requires no elaborate hyperparameter search. We find that coarse-grain DCAs are well suited for uncertainty-aware learning: modelwise DCA consistently outperforms all other uncertainty-aware learning methods for in-domain, distributional shift, and OOD scenarios; trunkwise DCA produces more diverse predictions while still delivering competitive performance w.r.t. deep ensemble, which is the best among existing baselines. Additionally, we discover that fine-grain DCA can produce a DCWA parameterization for the base network. DCWA improves the predictive performance compared to the standard baseline training. It has on-par performance with SWA but does not need custom learning rate scheduling and readjustment of batchnorm layers. Thus DCWA can be an attractive alternative to SWA that is easy to implement and deploy.

Similar to deep ensemble, DCA requires multiple times the computational budget to perform the joint training, compared to the standard baseline. While in this work we focused on the predictive power, we plan to improve the efficiency of DCA in future work. Apart from this, it can also be interesting to search for a variant that combines the benefits of both the fine-grain and coarse-grain DCAs, as currently DCWA is only possible for fine-grain DCAs, while good uncertainty estimations can only be attained via coarse-grain DCAs instead. Additionally, it is possible to combine DCA with other uncertainty-aware methods, which might further improve the performance. We plan to further investigate these aspects in future work.

## Acknowledgments and Disclosure of Funding

This work was supported by the Munich Center for Machine Learning (MCML) and by the ERC Advanced Grant SIMULACRON. The authors would like to thank Nikita Araslanov for proofreading and helpful discussions, as well as the anonymous reviewers and area chair for their constructive feedback.

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
