# OpenReview forum: "Deep Combinatorial Aggregation"
_NeurIPS.cc/2022/Conference — NeurIPS 2022 Accept_

### Official Review · Reviewer_tdmd · 2022-06-29

**Rating:** 4
**Confidence:** 4
**Soundness:** 2 fair
**Presentation:** 3 good
**Contribution:** 2 fair

**Summary:**

* **(1)** The authors introduce the deep combinatorial aggregation method, which could produce exponential ensemble model proposals by combining different parts of two neural networks.
* **(2)** A weight averaging strategy called DCWA is proposed to produce a new average model.
* **(3)** A consistency enforcing loss function for DCA is proposed to promote the similarity of different ensemble proposals.



**Questions:**

**However, I still have the following concerns and confusions:**
* **(Concerns 1)** Compared to other ensemble methods such as MCDropout [5] and SWA, the efficiency and complexity of this training method may increase significantly. Meanwhile, MCDropout[5] and other related works also could provide a variety of models without any additional computation cost. Compared with them, the method seems to have no advantages.
* **(Concerns 2)** Since only one model proposal is trained per iteration, I am concerned that the model is hard to converge when the number of ensemble proposals is large. Meanwhile, when the number of model proposals is large, the training time may increase significantly.
* **(Concerns 3)** As elaborated in Section 2.2, the model parameters are not decomposable in general. Therefore, a coarser multilayer trick is employed. However, this trick will reduce the number of model proposals.
* **(Concerns 4)** To help the DCA parameter training, predictive consistency is employed. However, as illustrated in Section 4, it has the risk of reducing the diversity of the model predictions.

**Limitations:**

This paper studies ensemble learning, a classic machine learning algorithm with almost no potential negative social impact.

**Strengths And Weaknesses:**

* **(Strengths 1)** The key motivation of this paper is similar and clear.
* **(Strengths 2)** The paper is well-organized and easy to follow.
* **(Strengths 3)** The potential limitations of the method have been well illustrated in the paper.

**However, I still have the following concerns and confusions:**
* **(Concerns 1)** Compared to other ensemble methods such as MCDropout [5] and SWA, the efficiency and complexity of this training method may increase significantly. Meanwhile, MCDropout[5] and other related works also could provide a variety of models without any additional computation cost. Compared with them, the method seems to have no advantages.
* **(Concerns 2)** Since only one model proposal is trained per iteration, I am concerned that the model is hard to converge when the number of ensemble proposals is large. Meanwhile, when the number of model proposals is large, the training time may increase significantly.
* **(Concerns 3)** As elaborated in Section 2.2, the model parameters are not decomposable in general. Therefore, a coarser multilayer trick is employed. However, this trick will reduce the number of model proposals.
* **(Concerns 4)** To help the DCA parameter training, predictive consistency is employed. However, as illustrated in Section 4, it has the risk of reducing the diversity of the model predictions.

---

> ### Author Response · Authors · 2022-08-02
> **Response to reviewer tdmd**
>
> Thank you for your feedback! We have created a paper revision to incorporate suggestions from the reviews.
>
> Before answering your specific concerns, we would like to first clarify some potential misunderstandings:
>
> - In concern 1, the claim "the method seems to have no advantages" is unjustified. Our experimental results clearly show that our method consistently outperforms existing baselines both in terms of accuracy and uncertainty estimation.
> - In concern 3, the "multilayer trick" corresponds to coarse-grained aggregation, and we do not require DCA models to always employ the "multilayer trick", i.e., constrain them to be coarse-grained. In fact, fine-grained (e.g., layerwise) DCA models are needed to ensure weight consistency and construct sensible DCWA parameterization. Section 2.3 provides a detailed discussion of this matter.
>
> To answer your specific concerns:
>
> >(Concerns 1) Compared to other ensemble methods such as MCDropout [5] and SWA, the efficiency and complexity of this training method may increase significantly. Meanwhile, MCDropout[5] and other related works also could provide a variety of models without any additional computation cost. Compared with them, the method seems to have no advantages.
>
> We respectfully disagree. Among other things, given the consistently superior results both in terms of predictive accuracy and uncertainty estimation, the claim "the method seems to have no advantages" is clearly unjustified. We have made it clear in our paper that our approach needs more (but still a reasonable amount of) computational budget compared to baselines like MCDropout or SWAG, but we are able to get higher quality predictions that consistently outperform existing baselines both in terms of accuracy and uncertainty estimation.
>
> > (Concerns 2) Since only one model proposal is trained per iteration, I am concerned that the model is hard to converge when the number of ensemble proposals is large. Meanwhile, when the number of model proposals is large, the training time may increase significantly.
>
> As explained in Section 5.3 paragraph "Number of component instances", DCA with more proposals can simply be trained with more epochs, and we observe no difficulty for the model to converge in practice. More proposals take longer to train, but it also improves the predictive performance.
>
> > (Concerns 3) As elaborated in Section 2.2, the model parameters are not decomposable in general. Therefore, a coarser multilayer trick is employed. However, this trick will reduce the number of model proposals.
>
> We have addressed this issue in Section 2.3. The granularity of DCA results in a trade-off between performance and prediction diversity. And one can alternatively use more DCA instances to increase the number of model proposals. Also, as mentioned previously, DCA models are not required to always employ the "coarser multilayer trick" which corresponds to a coarse-grain aggregation.
>
> > (Concerns 4) To help the DCA parameter training, predictive consistency is employed. However, as illustrated in Section 4, it has the risk of reducing the diversity of the model predictions.
>
> As discussed in Section 4, the consistency enforcing loss provides a design trade-off: it has the risk of reducing the predictive diversity, however it helps to better train the individual DCA components. In practice, we observe that its boosts the predictive performance of DCWA models and improves the uncertainty estimation of modelwise DCA.

---

> > ### Comment · Reviewer_tdmd · 2022-08-09
> > **Reply**
> >
> > After reading the author's reply, I still decided to keep my score fixed. The paper still has some unresolved concerns.
> >
> > 1. In principle, this method is very similar to MCDropout. It can be seen as dropping a part of the network in a larger network. MCdropout can be seen as dropping some scattered neurons in the neural network. In some extreme cases, the proposed method is consistent with MCDropout. Meanwhile, in principle, the proposed method has little advantage over other methods. For example, why does the proposed method perform better compared to deep ensemble or MCDropout?  What are the disadvantages of several completely independent networks (Deep ensembles) compared to such a combinable network?
> > 2. The proposed method brings little insight to the community. The constraints on consistency or diversity have been extensively explored in ensemble learning. Meanwhile, the proposed method can neither reduce the computational complexity nor provide a more reliable theoretical explanation.
> > 3. Consistency loss obviously reduces the diversity of the ensemble network.
> > 4. In practice, since only one model proposal is trained per iteration, I am concerned that the model is hard to converge when the number of ensemble proposals is large. Meanwhile, when the number of model proposals is large (more than 10), the training time may increase significantly.

---

> > > ### Author Response · Authors · 2022-08-09
> > > **RE: Reply**
> > >
> > > > 1.   In principle, this method is very similar to MCDropout. It can be seen as dropping a part of the network in a larger network. MCdropout can be seen as dropping some scattered neurons in the neural network. In some extreme cases, the proposed method is consistent with MCDropout. Meanwhile, in principle, the proposed method has little advantage over other methods. For example, why does the proposed method perform better compared to deep ensemble or MCDropout?
> > >
> > > We respectfully disagree with your claim "the proposed method has little advantage over other methods". As you have also mentioned, the proposed method outperforms deep ensemble and MCDropout. Designing a new approach that outperforms existing works for both predictive accuracy and uncertainty estimation is a main contribution of this paper. The proposed DCA model, together with the accompanying joint DCA training framework and the consistency enforcing loss, leads to superior performance that outperforms existing approaches.
> > >
> > > Regarding MCDropout, it is clearly different than our proposed DCA approach: MCDropout has the dropout rate hyperparameter and randomly zeros out activations, whereas DCA has no hyperparameter and uniformly samples weight copies to parameterize model proposals. Also, MCDropout only happens at neuron level, whereas DCA can be used with a full range of granularities from neuronwise to modelwise, as explained in Section 2.3 of the paper. We are unsure what you mean by "In some extreme cases, the proposed method is consistent with MCDropout", could you maybe elaborate on this claim?
> > >
> > > > 2.  The proposed method brings little insight to the community. The constraints on consistency or diversity have been extensively explored in ensemble learning. Meanwhile, the proposed method can neither reduce the computational complexity nor provide a more reliable theoretical explanation.
> > >
> > > We respectfully disagree with your conclusion "The proposed method brings little insight to the community". The focus of this work is to propose a new method that can achieve superior performance both for predictive accuracy and uncertainty estimation, not on the theoretical analysis of ensemble methods, nor on making deep ensemble more efficient. We believe our work makes non-trivial contributions to the community, and the other reviewers (dyii and GFCL) seem to agree.
> > >
> > > "The constraints on consistency or diversity have been extensively explored in ensemble learning.": Could you provide us with more concrete references so we could include them in the revision when appropriate? Thank you in advance!
> > >
> > > > 3.  Consistency loss obviously reduces the diversity of the ensemble network.
> > >
> > > We believe you are reiterating Concern 4 in your review which has already been addressed in our rebuttal. Do you have any questions regarding our response? Maybe you could elaborate on this point so we might answer your more specific questions?
> > >
> > > > 4.  In practice, since only one model proposal is trained per iteration, I am concerned that the model is hard to converge when the number of ensemble proposals is large. Meanwhile, when the number of model proposals is large (more than 10), the training time may increase significantly.
> > >
> > > We believe that you are reiterating Concern 2 in your review which has already been addressed in our rebuttal. Do you have any questions regarding our response?  Maybe you could elaborate on this point so we might answer your more specific questions?

---

> > > > ### Comment · Reviewer_tdmd · 2022-08-10
> > > > **Reply**
> > > >
> > > > Thanks for your response, I would like to know what are the advantages of the proposed method compared to Deep Ensemble or other ensemble methods?

---

### Official Review · Reviewer_dyii · 2022-07-11

**Rating:** 6
**Confidence:** 3
**Soundness:** 2 fair
**Presentation:** 3 good
**Contribution:** 3 good

**Summary:**

This paper proposes an effective ensembling technique for deep neural networks. The key idea is sharing the intermediate representation spaces among ensemble components. If we interpret deep neural networks as a chain of non-linear transformations mapping one representation space into another, this paper attempts to ensemble such intermediate mappings (DCA; Deep Combinatorial Aggregation). Furthermore, the authors suggest a weight averaging strategy compressing the proposed combinatorial ensemble to reduce the inference cost (DCWA; Deep Combinatorial Weight Averaging).

* * *
POST-REBUTTAL EDIT: Thank you for your time on the rebuttal. One remaining concern is the lack of large-scale experiments. Nevertheless, I believe the presented results (i.e., outperforming deep ensembles for in-domain, common corruptions, and out-of-distribution settings) are sufficient to verify the proposed method can benefit the community. Accordingly, I raised my score to 6.

**Questions:**


- How are the weights initialized for DCA and DCWA? In the case of DCWA, ensemble components should have nearly identical weight values to be averaged, and the same should probably go for the initial weights.
- In Table 3, why does CEL improve Modelwise DCA? It seems counter-intuitive; CEL reduces the predictive diversity, which can worsen the ensemble performance (as stated in lines 188-194). Perhaps CEL improves the performance of each individual ensemble member while decreasing the diversity among ensemble members. To check this, why don't we measure the classification performance of each individual?
- I also suggest clarifying that the improvement of Consistency Enforcing Loss (CEL) is distinct from regularization effects due to the smoothed label; the objective of CEL, defined in Equation 4, can be interpreted as Knowledge Distillation or Label Smoothing [Yuan et al., 2021]. I believe that further comparison with "NLL + Label Smoothing" will enhance the arguments in lines 160-178 and 295-302.
- Experimental evidence on large-scale datasets (e.g., ResNet-50 on ImageNet-1k) will be strong support for the paper, but I know that there are computational constraints to do so. One compromise is to start DCA (or DCWA) training from the pre-trained checkpoint (e.g., provided by torchvision); SWA [Izamilov et al., 2018] and SWA-Gaussian [Maddox et al., 2019] also provided ImageNet results in this way.

1. Yuan et al., Revisiting Knowledge Distillation via Label Smoothing Regularization, 2021.
2. Izmailov et al., Averaging Weights Leads to Wider Optima and Better Generalization, 2018.
3. Maddox et al., A Simple Baseline for Bayesian Uncertainty in Deep Learning, 2019.

**Limitations:**

This paper described the limitations of the work in Section 6; “Similar to deep ensemble, DCA requires multiple times the computational butdge to perform the joint training, compared to the standard baseline”.

**Strengths And Weaknesses:**

It is noteworthy that the proposed method explicitly verifies the benefits of ensembling intermediate mappings of deep neural networks. Although some of the existing Bayesian Neural Networks (BNNs) also can be interpreted in this way, further analysis of granularity provided in the paper makes distinct contributions.

I especially like the variety of experiments provided in the paper, including in-domain classification performance, robustness on common corruptions, and predictive uncertainty on out-of-distribution examples. Such experimental results clarify the benefits of the proposed ensembling method.

Nevertheless, there isn’t enough experimental evidence to fully accept the superiority of the proposed DCA and DCWA methods. Experiments for PreResNet-20 on CIFAR-10 and SVHN, provided in the paper, are limited in terms of scalability and generality. It is necessary to secure diversity in datasets and network structures.

---

> ### Author Response · Authors · 2022-08-02
> **Response to reviewer dyii**
>
> Thank you for your feedback!  We have created a paper revision to incorporate suggestions from the reviews.
>
> First, we would like to address your main concern about the diversity of datasets and network structures used in our experiments:
>
> - In terms of datasets, we believe the experiments can clearly demonstrate the effectiveness of DCA and DCWA. We use standard baselines (CIFAR/SVHN, CIFAR-C) for evaluation and conduct thorough experiments in in-domain, distributional shift, and OOD scenarios for uncertainty estimation. Due to limited computational resources, we could not run the full Imagenet/Imagenet-C experiments ourselves. However, we will release our source code and welcome others to test on other datasets.
>
> - In terms of network structures, we use ResNet which is a standard reference model used in the majority of existing related works, and our approach does not depend on any specific choice of network architecture. Moreover, to address your concern about the diversity of network structures, we have conducted additional experiments using VGG and DenseNet architectures and the results are included in Appendix D of the paper revision. Again we observe consistent best results both in terms of accuracy and uncertainty estimation.
>
> To answer your specific questions:
>
> > How are the weights initialized for DCA and DCWA? In the case of DCWA, ensemble components should have nearly identical weight values to be averaged, and the same should probably go for the initial weights.
>
> DCA and DCWA use the default LeCun random initialization from PyTorch and all copies start with the same initial weight (using different initialization for each copy yields similar performance). The results of DCWA have non-zero standard deviations, which suggests that the weights are not identical.
>
> > In Table 3, why does CEL improve Modelwise DCA? It seems counter-intuitive; CEL reduces the predictive diversity, which can worsen the ensemble performance (as stated in lines 188-194). Perhaps CEL improves the performance of each individual ensemble member while decreasing the diversity among ensemble members. To check this, why don't we measure the classification performance of each individual?
>
> As we have discussed in the paper, CEL takes inspiration from knowledge distillation / collaborative learning and can help the DCA training because it improves the performance of individual model copies. Following your suggestion, we conduct additional experiments to study individual models (see Appendix C of the revision). We observe that individual models from CEL training have reduced diversity (lower standard deviations) and better results, which matches our discussion in the paper. Thank you for your suggestion!
>
> > I also suggest clarifying that the improvement of Consistency Enforcing Loss (CEL) is distinct from regularization effects due to the smoothed label; the objective of CEL, defined in Equation 4, can be interpreted as Knowledge Distillation or Label Smoothing [Yuan et al., 2021]. I believe that further comparison with "NLL + Label Smoothing" will enhance the arguments in lines 160-178 and 295-302.
>
> There is a clear difference between CEL and label smoothing. Intuitively, CEL encourages predictions from different DCA model proposals to be consistent, whereas label smoothing augments the ground-truth output data and encourages predictions to have a predefined level of uncertainty determined by the smoothing factor.
>
> > Experimental evidence on large-scale datasets (e.g., ResNet-50 on ImageNet-1k) will be strong support for the paper, but I know that there are computational constraints to do so. One compromise is to start DCA (or DCWA) training from the pre-trained checkpoint (e.g., provided by torchvision); SWA [Izamilov et al., 2018] and SWA-Gaussian [Maddox et al., 2019] also provided ImageNet results in this way.
>
> As mentioned previously, we have limited computational resources and unfortunately could not run the full Imagenet/Imagenet-C experiments ourselves. Finetuning pre-trained models works well for SWAG, however this is a particular case that suits the SWAG approach. Finetuning from a pre-trained model does not fully explore the loss landscape, and for ensemble-type approaches, it is known to produce suboptimal results  (compare e.g., deep ensemble with snapshot ensemble for the same number of models [1]). It is therefore unsurprising that we also observe empirical performance deterioration when finetuning pre-trained models with DCA.
>
> [1] Gao Huang et. al. Snapshot Ensembles: Train 1, Get M for Free. ICLR 2017

---

> > ### Comment · Reviewer_dyii · 2022-08-07
> > **RE: Response to reviewer dyii (1/2)**
> >
> > Thank you for your clarification. Here are my comments on specific questions.
> >
> > > First, we would like to address your main concern about the diversity of datasets and network structures used in our experiments:
> > >
> > > * In terms of datasets, we believe the experiments can clearly demonstrate the effectiveness of DCA and DCWA. We use standard baselines (CIFAR/SVHN, CIFAR-C) for evaluation and conduct thorough experiments in in-domain, distributional shift, and OOD scenarios for uncertainty estimation. Due to limited computational resources, we could not run the full Imagenet/Imagenet-C experiments ourselves. However, we will release our source code and welcome others to test on other datasets.
> > >
> > > * In terms of network structures, we use ResNet which is a standard reference model used in the majority of existing related works, and our approach does not depend on any specific choice of network architecture. Moreover, to address your concern about the diversity of network structures, we have conducted additional experiments using VGG and DenseNet architectures and the results are included in Appendix D of the paper revision. Again we observe consistent best results both in terms of accuracy and uncertainty estimation.
> >
> > > > Experimental evidence on large-scale datasets (e.g., ResNet-50 on ImageNet-1k) will be strong support for the paper, but I know that there are computational constraints to do so. One compromise is to start DCA (or DCWA) training from the pre-trained checkpoint (e.g., provided by torchvision); SWA [Izamilov et al., 2018] and SWA-Gaussian [Maddox et al., 2019] also provided ImageNet results in this way.
> > >
> > > As mentioned previously, we have limited computational resources and unfortunately could not run the full Imagenet/Imagenet-C experiments ourselves. Finetuning pre-trained models works well for SWAG, however this is a particular case that suits the SWAG approach. Finetuning from a pre-trained model does not fully explore the loss landscape, and for ensemble-type approaches, it is known to produce suboptimal results (compare e.g., deep ensemble with snapshot ensemble for the same number of models [1]). It is therefore unsurprising that we also observe empirical performance deterioration when finetuning pre-trained models with DCA.
> >
> > As the authors stated, there can be problems with the fine-tuning approach for DCA (or DCWA) on ImageNet. The most apparent way is training from scratch, but I know the difficulty of conducting large-scale experiments. Nevertheless, I don't want to give a low score just because there is no large-scale experiment. Again, the various experiments provided in the paper sufficiently clarify the benefits of the proposed ensembling method and still potentially benefit the community. Moreover, additional experiments with other architectures (VGGNet and DenseNet) appear convincing to me.

---

> > > ### Comment · Reviewer_dyii · 2022-08-07
> > > **RE: Response to reviewer dyii (2/2)**
> > >
> > > > > In Table 3, why does CEL improve Modelwise DCA? It seems counter-intuitive; CEL reduces the predictive diversity, which can worsen the ensemble performance (as stated in lines 188-194). Perhaps CEL improves the performance of each individual ensemble member while decreasing the diversity among ensemble members. To check this, why don't we measure the classification performance of each individual?
> > > >
> > > > As we have discussed in the paper, CEL takes inspiration from knowledge distillation / collaborative learning and can help the DCA training because it improves the performance of individual model copies. Following your suggestion, we conduct additional experiments to study individual models (see Appendix C of the revision). We observe that individual models from CEL training have reduced diversity (lower standard deviations) and better results, which matches our discussion in the paper. Thank you for your suggestion!
> > >
> > > Thank you for accepting my proposal. I believe such an analysis of the proposed CEL makes the paper solid. However, using the standard deviations of evaluation metrics is not convincing; it is hard to say a set of members who have a higher standard deviation on individual performance are diverse. For instance, in the following binary classification situation, (1) both members #1 and #2 achieve a classification accuracy of 50% (i.e., zero standard deviation), but (2) they are definitely diverse. Since the evaluation metrics (including ACC, NLL, and ECE) are averaged over all test examples, the standard deviation over such metrics cannot identify whether each ensemble member makes different predictions on each test example.
> > >
> > > | | x1 | x2 | x3 | x4 | x5 | x6 | x7 | x8 |
> > > | :-: | :-: | :-: | :-: | :-: | :-: | :-: | :-: | :-: |
> > > | True Label | 0 | 0 | 0 | 0 | 1 | 1 | 1 | 1 |
> > > | Prediction of member #1 | 0 | 0 | 1 | 1 | 1 | 1 | 0 | 0 |
> > > | Prediction of member #2 | 1 | 1 | 0 | 0 | 0 | 0 | 1 | 1 |
> > >
> > > To make convincing arguments on the diversity of members (with/without CEL), I would like to suggest explicitly presenting the diversity metrics instead of the standard deviation of evaluation metrics. For instance, we can use the pairwise Kullback-Leibler divergence for the classification problem; (1) we first measure the per-example diversity metrics $D_1(x) = \frac{\sum_{i=1}^{M} \sum_{j=1}^{M} \text{KL}[p_i(x), p_j(x)]}{M(M-1)}$, and then (2) compute the average over test examples, i.e., $D_2 = \frac{1}{N} \sum_{n=1}^{N} D(x_n)$. Here,
> > > * $x_n$ denotes the $n$th test examples, for $n=1,...,N$,
> > > * $p_m$ denotes the $m$th ensemble members (which output categorical probabilities for a given input $x$), for $m=1,...,M$.
> > >
> > > Aside from this, there exist several ways to quantify ensemble diversity. For instance, Abe et al. (2022) contain the paragraph "Metrics for ensemble diversity," which summarizes two common metrics for measuring ensemble diversity. It will be nice to see what happens to the diversity of the ensemble members with these appropriate diversity metrics. For now, I will maintain my current score.
> > >
> > > ---
> > > (Abe et al., 2022) Deep Ensembles Work, But Are They Necessary? arXiv: 2202.06985

---

> > > > ### Author Response · Authors · 2022-08-08
> > > > **Reply**
> > > >
> > > > Although the edge case you showed does not happen in practice (we observe non-zero stds), we agree with you that reporting proper diversity metrics provides more convincing evidence.
> > > >
> > > > To summarize, there are three diversity metrics you have mentioned: pairwise KL divergence, classwise variance, and Jensen-Shannon divergence (The last two are from Abe et al. (2022)). We have computed all the metrics for our runs and the results are summarized as follows:
> > > >
> > > > |   | Pairwise KL | Variance | JS divergence |
> > > > |---|---|---|---|
> > > > |With CEL| $0.0052 \pm 0.0002$ | $0.0007 \pm 0.0000$ | $0.0020 \pm 0.0001$ |
> > > > | No CEL | $0.0077 \pm 0.0002$ | $0.0011 \pm 0.0000$ | $0.0031 \pm 0.0001$ |
> > > >
> > > > We see that no matter which metric we use, we reach the same conclusion discussed in the paper: CEL reduces the diversity of the individual predictions.
> > > >
> > > > We have made a new revision that incorporates these additional results. Thank you for the suggestion and the references!
> > > >
> > > > _Edit: there was an error where the Pairwise KL is computed two times bigger, now corrected._

---

> > > > > ### Comment · Reviewer_dyii · 2022-08-09
> > > > > **Post-rebuttal comments**
> > > > >
> > > > > Thank you for your time on the rebuttal. One remaining concern is the lack of large-scale experiments. Nevertheless, I believe the presented results (i.e., outperforming deep ensembles for in-domain, common corruptions, and out-of-distribution settings) are sufficient to verify the proposed method can benefit the community. Accordingly, I raised my score to 6.

---

### Official Review · Reviewer_GFCL · 2022-07-18

**Rating:** 6
**Confidence:** 4
**Soundness:** 3 good
**Presentation:** 4 excellent
**Contribution:** 3 good

**Summary:**

The paper proposes a new method called Deep Combinatorial Aggregation (DCA) which is a family of approaches for ensembling deep neural network models, often with weight-sharing. DCA trains multiple different versions of each neural network "component" (with different initializations), then samples from all possible combinations of the components at test-time to generate an empirical distribution of predictions. Among the "components" considered by the paper are layers, trunks (i.e., residual blocks), and complete models. The paper shows that the uncertainty estimates generated by DCA are comparable, or better than, existing methods such as MC-Dropout across CIFAR-10 and SVHN datasets, for in-domain, distribution shift, and out-of-distribution settings. The authors measure uncertainty via negative log-likelihood (NLL), Brier score, and ECE. Furthermore, the paper also proposes a method called deep combinatorial weight averaging (DCWA) for merging the models from a DCA ensemble into a single model.

**Questions:**

**Q1. Would it be appropriate to consider DCA to be generalization of MC-Dropout?** \
Specifically, it seems to me that neuronwise DCA is essentially MC-Dropout. Is my understanding correct?

**Q2. How is modelwise DCA different from deep ensemble?** \
The paper writes:
> Note that modelwise DCA is similar to deep ensemble, except that its model copies are trained with distinct data within each training epoch, ...

Does "its" refer to modelwise DCA or deep ensemble? In either case, I don't fully understand what you mean by "model copies are trained with distinct data within each training epoch."

**Q3. Why do the in-domain results not include layerwise DCA models?**

**Limitations:**

My main concern about DCA's limitations is how it scales to larger datasets/models and other data modalities. The authors do not seem to discuss this limitation.

**Strengths And Weaknesses:**

## Strengths

**S1. DCA generalizes the notion of using stochastic networks for predictive uncertainty estimates** \
Assuming my understanding to be correct (see Q1), DCA generalizes MC-Dropout to other stochastic model components, including layers, trunks, and even entire models. Furthermore, like dropout, DCA proposes a method (DWCA) to merge the individual models into a single model. This perspective is helpful in understanding ways to leverage model stochasticity for empirical uncertainty quantification.

**S2. Experiments show strength of DCA** \
The reasonably comprehensive set of experiments show that DCA outperforms (or at least matches) the performance of existing methods such as MC-Dropout and deep ensemble.

## Weaknesses

**W1. Unclear justification for consistency enforcing loss (CEL)** \
While the experimental evidence suggests that CEL is helpful, I struggle to understand why such an approach should be beneficial. It seems to go against the intuition that to get good uncertainty estimates, one should train diverse and independent models, instead of explicitly encouraging their predictions to be similar (which also breaks independence).

**W2. Lacking clarity in OOD experiments** \
How are the ROC curves in Figure 3 generated? Namely, how are you using the models trained on multi-class classification to perform outlier detection (binary classification)? Some more description is warranted.

**W3. Missing downstream tasks** \
While the results on uncertainty estimates are impressive, I would hope to see a discussion of how much the gain in uncertainty estimation performance affects downstream tasks (e.g., active learning or Bayesian optimization).

**W4. Missing discussion of scalability** \
How does the DCA approach scale to larger datasets or other data modalities? Would similar approaches work on, e.g., transformer models in NLP tasks?

---

> ### Author Response · Authors · 2022-08-02
> **Response to reviewer GFCL**
>
> Thank you for your feedback!  We have created a paper revision to incorporate suggestions from the reviews.
>
> We would like to address your concerns and questions in the following:
>
> - W1. Unclear justification for consistency enforcing loss (CEL)
>
> CEL takes inspiration from knowledge distillation / collaborative learning and can help the DCA training because it improves the performance of individual model copies. We have conducted additional experiments to study individual models in Appendix C of the revision. There we observe that individual models from CEL training have reduced diversity (lower standard deviations) and better results, which matches our discussion in the paper.
>
> - W2. Lacking clarity in OOD experiments
>
> We follow the standard experimental setting for the OOD benchmark and have provided the reference [1] in the paper. Specifically, we construct the ROC curves based on the probability of the predicted class (i.e. maximum value of the softmax output). In an uncertainty-aware model, this probability should be higher for in-domain samples and lower for OOD ones, indicating high uncertainty.
>
> - W3. Missing downstream tasks
>
> We fully agree that investigating DCA for downstream tasks is interesting to consider. This said, given the page limit and the focus of the paper, an in-depth study on applying DCA to downstream tasks like active learning or Bayesian optimization would be beyond the current scope and better addressed in future work.
>
> - W4. Missing discussion of scalability
>
> Since DCA does not have any domain-specific components, we expect it to work similarly well beyond the established uncertainty estimation benchmarks, on which we focused in this work. In terms of computational complexity, the DCA approach, like other common deep learning approaches, scales linearly w.r.t. dataset and network size. DCA may also be applied to transformer models. We will include additional NLP-related experiments for camera-ready. Thank you for the suggestion!
>
> - Q1. Would it be appropriate to consider DCA to be generalization of MC-Dropout?
>
> Although MC dropout and neuronwise DCA share the similar idea of getting subsampled models, they correspond to different approaches. MC dropout has the dropout rate hyperparameter and randomly zeros out activations, whereas DCA has no hyperparameter and uniformly sample weight copies to parameterize model proposals.
>
> - Q2. How is modelwise DCA different from deep ensemble?
>
> “its” refers to modelwise DCA, we have revised it to avoid confusion. During joint DCA training, each data sample would be only used to update DCA copies yielded from the random selection process. This means that each model sampled by DCA will only see a different subset of the training data during each epoch. Deep ensembles, on the other hand, use the full training set to train their models during each epoch.
>
> - Q3. Why do the in-domain results not include layerwise DCA models?
>
> In-domain results for DCA models with different levels of granularity (including layerwise DCA) are reported and analyzed in ablation studies (Section 5.3, Figure 4). For the comparative experiments on uncertainty-aware learning in Section 5.2, we focus on trunkwise and modelwise DCA variants which have the best uncertainty estimation results.
>
> [1] D. Hendrycks and K. Gimpel. A baseline for detecting misclassified and out-of-distribution examples in neural networks. ICLR 2017

---

### Meta-Review · Area_Chair_Jq8a · 2022-08-26

**Recommendation:** Accept
**Confidence:** Certain

**Metareview:**

The work proposes an ensembling technique for improved uncertainty and OOD predictive performance. The idea is to use multiple permutations of a certain component (e.g., layers, residual blocks) for both training and testing. Like efficient ensembles such as BatchEnsemble, most parameters are shared across ensemble members and a component is used multiple times depending on its selection for a given permutation in the combinatorial space. Overall, this results in an ensemble over a wider variety of networks, with a higher diversity than if one used completely independent networks for each ensemble member. There are further variants for efficiency such as "Deep combinatorial weight averaging".

The idea is conceptually simple and only requires the addition of a training objective to enforce consistency. The experiments are also quite thorough.

I agree with two of the three reviewers, disagreeing with Reviewer tdmd. One of the main concerns is that while the method outperforms baselines like MC Dropout, there's a lack of a theoretical explanation. I believe this can be helpful but is not a necessity, particularly when theory for efficient ensembles of neural networks is quite a difficult setting to study formally without sweeping assumptions. Other concerns like efficiency/complexity compared to more efficient methods like MC Dropout, SWA, and BatchEnsemble; and training time and convergence difficulties are adequately addressed in the rebuttal.

In related work, I would recommend discussing recent works that have also examined improving the diversity of ensembles for improved uncertainty/robustness, particularly those that have proposed a diversity penalty. To name a few: [1], [2], [3].

[1] A Diversity-Penalizing Ensemble Training Method for Deep Learning. Wentao Zhang, Jiawei Jiang, Yingxia Shao, Bin Cui https://arxiv.org/abs/2112.13316

[2] Hyperparameter Ensembles for Robustness and Uncertainty Quantification. Florian Wenzel, Jasper Snoek, Dustin Tran, Rodolphe Jenatton https://arxiv.org/abs/2006.13570

[3] Learning under Model Misspecification: Applications to Variational and Ensemble methods. Andres R. Masegosa https://arxiv.org/abs/1912.08335

**Award:**

No

---

### Decision · Program_Chairs · 2022-09-14

Accept